# Modeling Spatiotemporal Heterogeneity in Earth Science Machine Learning: An End-to-End Approach

## Abstract

In Earth sciences, unobserved factors often lead to spatially nonstationary distributions, causing relationships between features and targets to vary across locations. Traditional tabular machine learning methods struggle to effectively model this spatial heterogeneity. While approaches like Geographically Weighted Regression (GWR) capture local variations, they often miss global patterns, overfit local noise, and lack the ability to model temporal changes in spatial heterogeneity. Our research aims to model spatiotemporal heterogeneity. To achieve this, we propose an end-to-end approach that fits the entire dataset to capture global patterns, while designing the model as a conditional generative framework to learn sparse spatial heterogeneity, mitigating overfitting through localized condition sharing. Our method involves four key steps: constructing a spatiotemporal graph, encoding tabular features, aggregating spatial heterogeneity node embeddings via graph convolutions, and decoding with spatial condition vectors for location-specific predictions. We validate our approach by predicting vegetation gross primary productivity (GPP) using global climate and land cover data (2001–2020). Trained on 50M samples and tested on 2.8M, our model achieves an RMSE of 0.836, outperforming GWR (2.149), LightGBM (1.063) and TabNet (0.944). Visual analysis of the learned node embeddings reveals clear spatial heterogeneity patterns and their temporal dynamics.

## 1 Introduction

In Earth science, tabular machine learning is widely used to model environmental and geographical relationships, such as predicting climate change impacts on vegetation (Lu et al., 2024) and understanding tropical cyclones' effects on precipitation (Qin et al., 2024). Accurate modeling is crucial for reliable environmental predictions.

However, most tabular machine learning methods assume unordered samples, raising questions about their applicability to all Earth science problems. While a global mapping requires all influencing factors to be known, many factors in Earth science, such as soil nutrients, microbial activity, and biodiversity, are difficult to measure, leading to incomplete information. This introduces a significant challenge: the spatial distribution of missing variables is often non-stationary, meaning that the relationship between the remaining features and the target variable changes with spatial location (Fotheringham et al., 2009). For example, the relationship between temperature and vegetation carbon accumulation rates may vary across regions due to differences in species, soil quality, and altitude. Current models capture common patterns but fail to address spatial heterogeneity, highlighting the need for better methods to model spatial variability.

One solution is to use local models like Geographically Weighted Regression (GWR) (Fotheringham et al., 2009), which adjusts coefficients based on location to capture spatial variability. However, GWR lacks temporal modeling, limiting its ability to capture the evolution of spatial heterogeneity. To address GWR's temporal limitation, Geographically and Temporally Weighted Regression (GTWR) (Fotheringham et al., 2015) was developed, using spatiotemporal metrics to model variability. However, GTWR struggles with nonlinearity, prompting the development of hybrid methods like Geographically and Temporally Neural Network Weighted Regression (GTNNWR) (Wu et al.,

2021), which uses a Spatiotemporal Proximity Neural Network (STPNN) to model nonlinear spatiotemporal heterogeneity.

Despite these advancements, current methods still fit spatial weights locally, based on neighborhood samples, leading to several challenges: 1) Learning objective: Local models may miss common patterns across regions by focusing too much on local variability; 2) Model complexity: Fitting spatial weights for each location can result in a highly dense parameter space, making the model prone to overfitting local noise; 3) Computational efficiency: In Earth science, datasets are often massive, and methods with computational complexity proportional to sample size may struggle with large-scale data.

Our study aims to learn spatiotemporal heterogeneity, aligning with previous methods' objectives. To address the limitations of existing research, we draw inspiration from the success of end-to-end learning in computer science, particularly the DETR model (Carion et al., 2020). We propose a unified optimization process that leverages an end-to-end learning framework to capture the spatiotemporal heterogeneity of variable relationships across the entire sample space. Additionally, instead of using explicit geographically and temporally weighted models, we propose a conditional generative model with local parameter sharing, reducing the risk of overfitting due to dense spatiotemporal weights. Our approach aims to effectively model the dynamic and spatially varying relationships between variables in Earth science data.

Building on this idea, we have developed method, which employs graph neural networks (GNNs) (Wu et al., 2020) to implicitly learn mappings with spatiotemporal heterogeneity from Earth science data. Our method consists of four key components: First, in the preprocessing stage, we cluster the global land grid and map the cluster centers to spherical coordinates, using the K-nearest neighbors algorithm to construct a spatial adjacency graph. Each cluster category shares a spatial condition vector during decoding. In the representation learning and prediction stages, we employ a tabular feature encoding module and a spatial heterogeneity encoding module to encode the tabular data features and spatial heterogeneity conditions, respectively. The decoding module uses this encoded information to predict the target variables. The tabular feature encoding module uses linear self-attention over two dimensions to simultaneously capture the attribute features of the tabular data and their temporal dynamics. The spatial heterogeneity module aggregates node embeddings using a spatiotemporal GCN (Graph Convolutional Network (Kipf & Welling, 2016)), producing spatial condition vectors that describe the spatial heterogeneity at each location. Finally, the decoding module uses the spatial condition vectors as target vectors and the tabular feature encodings as memory vectors, applying a transformer decoder to generate predictions.

To validate our approach, we created the Climate2GPP dataset, using the ERA5 climate dataset (Muñoz-Sabater et al., 2021), the MCD12C1.061 MODIS Land Cover dataset (Friedl & Sulla-Menashe, 2022), and the PML_V2 0.1.7 GPP dataset (Zhang et al., 2019). Spanning from 2001 to 2020 with an 8-day temporal resolution, this dataset includes approximately 50 million samples for training and 2.8 million samples for testing. Our method achieved an RMSE of 0.836 on the test set, significantly outperforming GWR (RMSE 1.937), classical tabular machine learning methods like LightGBM Large (RMSE 1.063) and deep learning methods like TabNet (RMSE 0.944).

We also analyzed the GNN's node embeddings visually, observing spatial distribution patterns that help us understand the spatial heterogeneity of variable relationships and their temporal evolution.

## 2 RELATED WORKS

**Tabular Machine Learning and Geographically Weighted Models:** Tabular machine learning methods have been widely applied in Earth science for tasks such as predicting environmental changes and understanding geographical phenomena. These methods, including popular algorithms like LightGBM (Ke et al., 2017) and XGBoost (Chen & Guestrin, 2016), are designed under the assumption that samples are independent and identically distributed, which limits their applicability in scenarios with spatial dependencies. While geographically weighted models, such as GWR (Fotheringham et al., 2009) and GTWR (Fotheringham et al., 2015), have been introduced to address spatial heterogeneity by adjusting coefficients locally, they face significant limitations. GWR models fail to capture temporal evolution, and while GTWR extends this capability, both models struggle with nonlinearity and exhibit high computational complexity when applied to large datasets. GWR-RF

(Wang et al., 2024) combines GWR with Random Forest for nonlinearity but still suffers from local overfitting and dense weight matrices. GNNWR (Du et al., 2020) balances global patterns and spatial variability through neural network-corrected coefficients but retains the complexity of dense spatial weights. GTNNWR (Wu et al., 2021) further incorporates spatiotemporal heterogeneity but remains limited by the need to fit local variables and dense spatiotemporal weights, making these models prone to overfitting and computationally inefficient in large-scale applications.

## 3 METHOD

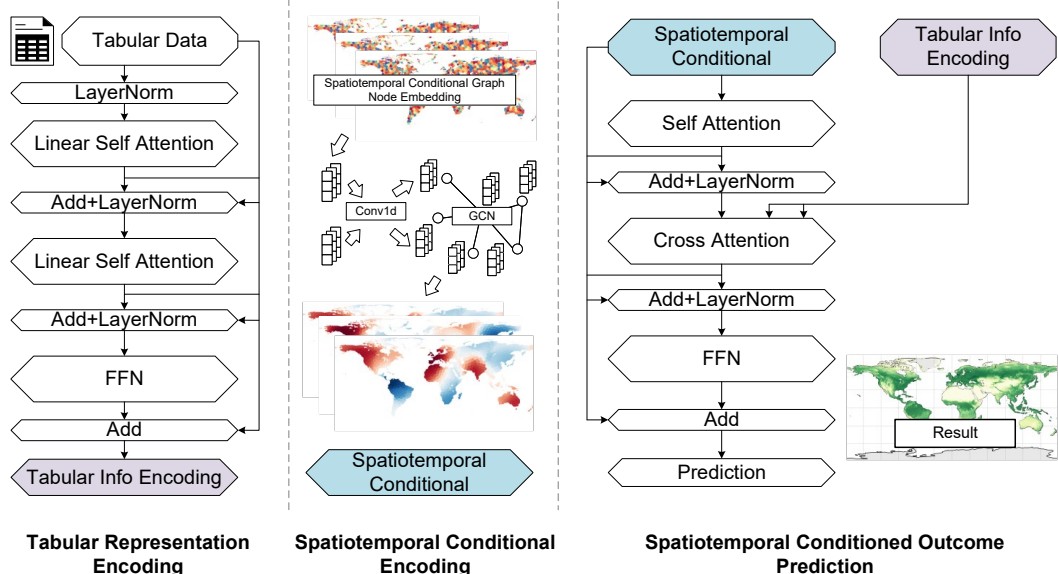

**Figure 1: Overall workflow.**

We address the problem of learning under spatiotemporal heterogeneity by framing it as a conditional generation task, where predictions are made based on data attributes conditioned on spatiotemporal contexts. Our approach focuses on four key issues: representing spatiotemporal conditions, encoding tabular attributes, generating predictions under these conditions, and ensuring end-to-end optimization. To represent the spatiotemporal conditions, we construct a graph where node embeddings are learned by aggregating local spatiotemporal information using graph convolutions. For encoding tabular attributes, we design a dual-attention transformer encoder that captures both temporal and feature-level dependencies. The prediction process utilizes a transformer decoder, where spatiotemporal conditions are treated as the target sequence and tabular data as the memory sequence. The entire framework, including learnable node embeddings, feature aggregation, and prediction modules, is optimized end-to-end through gradient descent.

### 3.1 SPATIOTEMPORAL CONDITIONAL GRAPH CONSTRUCTION

To capture the spatiotemporal heterogeneity, we propose a **Spatiotemporal Conditional Graph** (STCG). The STCG is defined as $G = (V, E)$, where $V$ is the set of nodes and $E$ is the set of edges. Each node $v_{i,t} \in V$ represents a spatiotemporal point $(\lambda_i, \phi_i, t)$, where $\lambda_i$ and $\phi_i$ are the longitude and latitude of node $i$, and $t$ is the time. Each node has an embedding $\boldsymbol{v}_{i,t}$ that captures the spatiotemporal condition at that point. The prediction for a spatiotemporal location is influenced by the embedding $\boldsymbol{v}_{i,t}$ of the corresponding node in the STCG. The construction of the STCG involves the following steps.

**Graph Node Generation:** To determine the geographical coordinates $(\lambda_i, \phi_i)$ of each node $v_{i,t} \in V$, we apply K-means clustering to the global land grid. This step reduces computational complexity by grouping spatial regions into clusters, where each cluster shares a common spatiotemporal condition. Specifically, the cluster centers $\boldsymbol{C} = \{\boldsymbol{c}_1, \boldsymbol{c}_2, \ldots, \boldsymbol{c}_k\}$ are determined by minimizing the

sum of squared distances between all spatial points and their nearest cluster centers:

$$C = \arg\min_{C} \sum_{p=1}^{n} \min_{j} \|(\lambda_p, \phi_p) - c_j\|^2$$

Here, $(\lambda_p, \phi_p)$ represents any spatial point $p$ in the global grid, and $n$ is the total number of such points. Each node $v_{i,t} \in V$ is then assigned the spatial coordinates of the corresponding cluster center: $(\lambda_i, \phi_i) = c_i$.

**Cyclic Graph Construction:** To ensure connectivity between the eastern and western hemispheres, we map the geographical coordinates of the cluster centers to spherical coordinates and construct the adjacency matrix $A$ using the K-nearest neighbors (KNN) method on the sphere. Specifically, for each cluster center $v_{i,t}$, we first project its geographical coordinates $(\lambda_i, \phi_i)$ (longitude and latitude) onto a 3D unit sphere using the following transformation:

$$x_i = \cos(\phi_i)\cos(\lambda_i), \quad y_i = \cos(\phi_i)\sin(\lambda_i), \quad z_i = \sin(\phi_i)$$

where $(x_i, y_i, z_i)$ represents the 3D spherical coordinates of node $v_{i,t}$. Using the spherical coordinates $p_i = (x_i, y_i, z_i)$, we compute the adjacency matrix $A$ of the graph by defining the $k$-nearest neighbors for each node $v_{i,t}$. The adjacency matrix $A$ is constructed as follows:

$$A_{i,j} = \begin{cases} 1, & j \in \arg\text{top-k}\,\|p_i - p_j\| \\ 0, & \text{otherwise} \end{cases}$$

Here, $p_i$ and $p_j$ are the 3D spherical coordinates of nodes $v_{i,t}$ and $v_{j,t}$, respectively. This approach ensures that the graph is cyclic, connecting locations on opposite sides of the globe, which is particularly important for capturing the circular nature of the Earth.

**Node Embedding Calculation:** We use Node2vec (Grover & Leskovec, 2016) to compute the initial embeddings for the nodes. For each time dimension $t$, we add a time embedding using the Rotational Position Embedding (RoPE) (Su et al., 2024) method: $v_{i,t} = \text{Node2vec}(i) + \text{RoPE}(t)$ where $v_{i,t}$ is the embedding of node $v_{i,t}$ at time $t$.

**Edge Weight Calculation:** The weight of each edge $w_{i,j}$ in the graph is computed using a log-Gaussian kernel, which incorporates two sequential normalization steps to effectively capture the similarity between cluster centers in a 3D space. The weight of each edge $w_{i,j}$ can be calculated by:

$$w_{i,j} = \begin{cases} \exp\left(-\dfrac{\left(1-\exp\left(-\frac{\|p_i - p_j\|}{\mu}\right)\right)^2}{2\sigma^2}\right), & \text{if } j \in \arg\text{top-k}\,\|p_i - p_j\| \\ 0, & \text{otherwise} \end{cases}$$

### 3.2 SPATIOTEMPORAL CONDITIONAL ENCODING

Building upon the Spatiotemporal Conditional Graph (STCG) construction, we propose a Spatiotemporal Conditional Encoding method to aggregate temporal and spatial information within the graph, effectively modeling the spatiotemporal interactions across different locations. This process aims to derive heterogeneous descriptive vectors for each spatiotemporal point, capturing the complex interdependencies in the data.

The Spatiotemporal Conditional Encoding can be decomposed into two main steps: temporal aggregation and spatial aggregation. For each node $v_{i,t}$ in the STCG, we update its embedding $v_{i,t}$ through these aggregation processes.

**Temporal Aggregation:** We first apply a 1D convolution operation along the time dimension to capture temporal dependencies. This can be formulated as:

$$V^{temp} = V * W^{time}$$

where $V$ is the matrix of node embeddings $v_{i,t}$, $W^{time}$ is the learnable temporal convolution kernel, and $*$ denotes the convolution operation.

**Spatial Aggregation:** Following the temporal aggregation, we employ a graph convolution operation to aggregate spatial information:

$$V^{final} = \sigma(D^{-\frac{1}{2}} A W D^{-\frac{1}{2}} V^{temp} H)$$

where $\boldsymbol{A}$ is the adjacency matrix, $W$ is the edge weight matrix $w_{i,j}$, $D$ is the degree matrix $d_i$, $H$ is the learnable weight matrix, and $\sigma$ is a non-linear activation function. The final embedding for each node $v_{i,t}$, incorporating both temporal and spatial information, can be expressed as:

$$\boldsymbol{v}_{i,t}^{final} = \sigma\left(\sum_{j \in \mathcal{N}(i)} \frac{1}{\sqrt{d_i d_j}} w_{i,j} H\left(\sum_{\tau=-k}^{k} w_\tau^{time} \cdot \boldsymbol{v}_{j,t+\tau}\right)\right)$$

where $\mathcal{N}(i)$ is the set of neighboring nodes of node $v_{i,t}$, $d_i$ and $d_j$ are the degrees of nodes $i$ and $j$, $w_{i,j}$ is the edge weight between nodes $v_{i,t}$ and $v_{j,t}$, and $w_\tau^{time}$ are the elements of the temporal convolution kernel $W^{time}$. By applying these operations sequentially, we obtain a rich representation $\boldsymbol{v}_{i,t}^{final}$ for each spatiotemporal point $v_{i,t}$, which encapsulates both local and global spatiotemporal dependencies.

### 3.3 TABULAR REPRESENTATION ENCODING

Our primary goal is to develop an efficient module that extracts both temporal representations (capturing seasonal variation patterns) and feature representations from Earth science tabular data. This process begins with a feature mixing step, where the input feature space is projected to $N$ features. Following this, rotational position encoding (RoPE) (Su et al., 2024) is applied to the temporal dimension to incorporate positional information. Finally, we utilize stacked Dual Attention (DA) modules to extract both temporal and feature-based dependencies.

**Dual Attention Mechanism:** Given an input tensor $X \in \mathbb{R}^{L \times D}$, where $L$ represents the sequence length (time dimension) and $D$ is the feature dimension, the DA module sequentially computes self-attention (Katharopoulos et al., 2020) across the temporal and feature dimensions.

First, temporal self-attention is applied across the time steps for each feature. Queries, keys, and values are computed as:

$$Q^{\text{temp}} = XW_Q^{\text{temp}}, \quad K^{\text{temp}} = XW_K^{\text{temp}}, \quad V^{\text{temp}} = XW_V^{\text{temp}}$$

The temporal attention output is then computed by applying the activation function $\phi(x) = \text{ELU}(x) + 1$ directly within the attention formula:

$$\text{Attn}^{\text{temp}} = \frac{\phi(Q^{\text{temp}})\left(\phi(K^{\text{temp}})^\top V^{\text{temp}}\right)}{\phi(Q^{\text{temp}})\left(\phi(K^{\text{temp}})^\top \mathbf{1}_L\right) + \epsilon}$$

Feature self-attention is then computed along the feature dimension using the same process. Residual connections are employed at each step to ensure gradient flow and model stability:

$$X^{\text{feat}} = X^{\text{temp}} + \text{Attn}^{\text{feat}}$$

**Feedforward Network:** Finally, a position-wise feedforward network is applied to each element:

$$Y = X^{\text{feat}} + \text{FFN}(X^{\text{feat}})$$

By employing both temporal and feature self-attention, followed by a feedforward network, this model captures rich representations from tabular Earth science data.

### 3.4 SPATIOTEMPORAL CONDITIONED OUTCOME PREDICTION

After encoding the tabular features and spatiotemporal heterogeneity conditions, we employ a transformer layer to decode the outcomes. Specifically, in the decoding stage, the spatio-temporal conditions are used as the target sequence, while the encoded tabular features serve as the memory sequence. Let $\mathbf{X} \in \mathbb{R}^{T \times d}$ represent the encoded tabular features with $T$ timesteps and $d$-dimensional feature embeddings, and $\mathbf{C} \in \mathbb{R}^{S \times d}$ represent the spatiotemporal conditions with $S$ spatio-temporal steps. The transformer decoder computes an output $\mathbf{Z}$ as follows:

$$\mathbf{Z} = \text{Decoder}(\mathbf{C}, \mathbf{X})$$

Finally, a linear layer is then applied to the decoder output to generate the final predictions $\hat{\mathbf{y}} \in \mathbb{R}^T$ for each timestep in the sequence:

$$\hat{\mathbf{y}} = \text{Linear}(\mathbf{Z})$$

## 4 EXPERIMENTS

### 4.1 EXPERIMENTAL SETUP

**Dataset:** To validate our approach, we created the Climate2GPP dataset. We used data from Google Earth Engine spanning January 1, 2001, to December 17, 2020. The data sources include:

- **ERA5-Land Daily Aggregated** (ECMWF): Global historical meteorological data aggregated every 8 days (Muñoz-Sabater et al., 2021).
- **MCD12C1.061 MODIS Land Cover Type** (NASA): Yearly global land cover changes at 0.05 degree resolution (Friedl & Sulla-Menashe, 2022).
- **PML_V2 0.1.7**: Global gross primary productivity (GPP) data aggregated every 8 days (Zhang et al., 2019).

From ERA5, we selected 26 climate parameters, with solar radiation, evaporation, and precipitation summed over 8-day periods, while the rest were averaged. GPP was similarly summed over 8-day intervals. Data from 2001 to 2019 was used for training (52M samples), and data from 2020 served as the test set (2.8M samples).

**Training Setting:** Our method is implemented in PyTorch 2.1.2 with CUDA 11.8. All features, except GPP, are normalized. The AdamW optimizer is used with a batch size of 256, an initial learning rate of 0.001, decayed to 0.0001 after 10 epochs, for a total of 20 epochs.

Comparison machine learning method are (KNN, Random Forest, XGBoost, LightGBM, CatBoost) using AutoGluon 1.1.1 (Erickson et al., 2020) with default hyperparameters. These models, along with deep learning comparisons (TabNet, ResNet, ExcelFormer, FFTransformer implemented in PyTorchFrame 0.2.3 (Hu et al., 2024)), are trained on RTX 4090 GPU, 64-core Intel Xeon Platinum 8352V and 120GB of RAM. All deep learning models use the same optimizer settings as our method.

### 4.2 RESULT COMPARISON

To validate the suitability of our proposed method for machine learning tasks in the Earth sciences, we conducted a comparative evaluation against a range of widely adopted machine learning baselines, including Random Forest (Breiman, 2001), XGBoost (Chen & Guestrin, 2016), CatBoost (Prokhorenkova et al., 2018), the LightGBM family (Ke et al., 2017), and KNN. Additionally, we compared our method with state-of-the-art tabular deep learning approaches, including TabNet (Arik & Pfister, 2019), ExcelFormer (Chen et al., 2024), ResNet (Gorishniy et al., 2021), and FTTransformer (Gorishniy et al., 2021). All models were trained using the complete dataset of 50 million samples to assess their scalability and performance on large-scale data. The prediction accuracy of each method was evaluated on the Climate2GPP test set for estimating the total gross primary productivity (GPP) for the year 2020, as detailed in the table (2).

Figure 2: Comparison of Different Methods

| Method | RMSE | $R^2$ |
|---|---|---|
| Tabular Machine Learning | | |
| LightGBM Large | 1.063 | 0.886 |
| KNeighborsDist | 1.093 | 0.879 |
| KNeighborsUnif | 1.096 | 0.879 |
| LightGBM | 1.108 | 0.876 |
| XGBoost | 1.124 | 0.872 |
| LightGBMXT | 1.126 | 0.872 |
| NeuralNetFastAI | 1.142 | 0.868 |
| CatBoost | 1.152 | 0.866 |
| RandomForestMSE | 1.182 | 0.859 |
| Tabular Deep Learning | | |
| TabNet | 0.944 | 0.901 |
| ExcelFormer | 1.001 | 0.878 |
| ResNet | 1.014 | 0.878 |
| FTTransformer | 1.158 | 0.850 |
| Our Method | | |
| Ours | **0.836** | **0.932** |

As shown in Table 2, the best-performing machine learning and deep learning methods on this task were LightGBM Large and TabNet, achieving RMSEs of 1.063 and 0.944, respectively. However, our proposed method outperformed both, achieving a lower RMSE of 0.836 and an $R^2$ of 0.932. These results underscore the superior capability of our approach in handling large-scale Earth science data. Moreover, they suggest that by accounting for the spatiotemporal heterogeneity in the relationships between independent and dependent variables, significantly improved predictive performance can be achieved for Earth science problems.

### 4.3 COMPARISON OF SPATIAL AND SPATIOTEMPORAL HETEROGENEITY METHODS

Furthermore, we aim to compare our method with other approaches that are capable of modeling spatial or spatiotemporal heterogeneity. Notably, the computational complexity of the GWR series methods is proportional to the number of spatial locations in the dataset. Moreover, GWR series methods require exactly one sample point per spatial location. Given these limitations, all experiments in this section were conducted on a smaller dataset. This setup ensures a fair comparison between our method and GWR, while also evaluating our method's fitting performance on a smaller dataset. Specifically, we uniformly sampled 6,000 grids from the land grid as training data, with each grid containing data from all available time points. Since GWR series methods require a one-to-one correspondence between samples and locations, we used the average data from every 8 days over 19 years as the training samples. For GWR (Fotheringham et al., 2009) and GNNWR (Du et al., 2020), which cannot model spatiotemporal heterogeneity, we fit separate weekly temporal models. For all models, we selected results from Weeks 1, 10, 20, 30, and 40, which are representative of different seasons, for comparison. The results are shown in Table 1.

Table 1: Comparison of Spatial and Spatiotemporal Heterogeneity Methods (RMSE / $R^2$)

| Method | Week-1 | Week-10 | Week-20 | Week-30 | Week-40 | Overall |
|--------|--------|---------|---------|---------|---------|---------|
| | Modeling Spatial Heterogeneity | | | | | |
| GWR | 1.990/0.434 | 2.097/0.424 | 2.184/0.592 | 2.060/0.506 | 1.958/0.429 | 2.149/0.534 |
| GNNWR | 0.871/0.891 | 1.066/0.851 | 1.330/0.855 | 1.178/0.838 | 0.835/0.896 | - |
| | Modeling Spatiotemporal Heterogeneity | | | | | |
| GTWR | 1.761/0.557 | 1.859/0.547 | 2.475/0.476 | 2.070/0.501 | 1.689/0.575 | - |
| Ours | **0.779/0.913** | **0.813/0.914** | **1.073/0.905** | **0.931/0.887** | **0.700/0.922** | 0.836/0.932 |

In addition, we visualized the spatial heterogeneity weights for Week 40 (using PCA to reduce the dimensionality of all variable weights to one dimension), as shown in Figure 3.

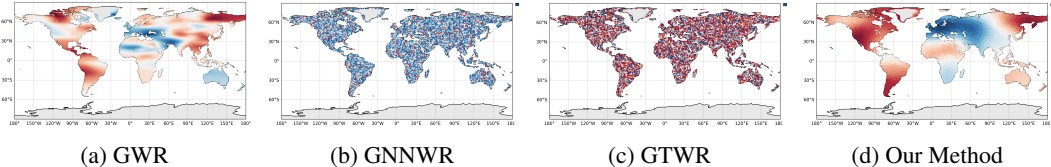

| (a) GWR | (b) GNNWR | (c) GTWR | (d) Our Method |
|---------|-----------|----------|----------------|

Figure 3: Visualization of spatial heterogeneity weights in week-40

As shown in Table 1 and Figure 3, compared to GWR and GTWR (Fotheringham et al., 2015), which can only fit linear relationships, methods that can model nonlinear relationships have a clear advantage in terms of RMSE. Compared to GTWR, the sparsity-based learning of spatiotemporal heterogeneity in our method significantly alleviates overfitting, and the learned spatiotemporal heterogeneity weights exhibit a smooth spatial distribution.

Table 2: Comparison of Spatial and Spatiotemporal Heterogeneity Methods

| Method | Train RMSE | Train $R^2$ | Test RMSE | Test $R^2$ | Generalization Gap |
|--------|-----------|-------------|-----------|------------|--------------------|
| GNNWR | 0.478 | 0.931 | 0.835 | 0.896 | 0.357 |
| Ours | 0.627 | 0.942 | 0.700 | 0.922 | 0.073 |

Finally, in comparison to GNNWR (Du et al., 2020), which also models nonlinear relationships, our method achieves better results because it can leverage the entire sample space (with different times and locations) to learn the common patterns across all times and locations in an end-to-end manner. Comparing the training and testing RMSEs (Table 2), GNNWR shows a training RMSE of 0.478 for Week 40, while the testing RMSE is 0.835, resulting in a difference of around 0.35. In contrast, our method yields a training RMSE of 0.627 and a testing RMSE of 0.700, with a difference of around 0.08. Additionally, compared to GNNWR, the spatiotemporal heterogeneity weights learned by our method are smoother. These findings demonstrate that the improvements in our method effectively mitigate local overfitting, a common issue when learning spatiotemporal heterogeneity.

## 4.4 ABLATION EXPERIMENT

We conducted an ablation study to evaluate the contribution of each module proposed in this paper. The baseline model is the FFTransformer (Gorishniy et al., 2021), which includes only an encoder (En) that processes feature dimensions without incorporating temporal or spatial information. As shown in Table 4, we systematically examined the effects of adding temporal decoding, spatiotemporal graph modeling, and our proposed enhancements to graph construction and feature extraction. The results, presented in terms of RMSE, demonstrate the impact of each module.

Figure 4: Ablation Study

| Method | RMSE |
|---|---|
| FFTransformer (En only) | 1.158 |
| En+De | 1.071 |
| En+De+GCN | 0.893 |
| En+De+GCN+EG | 0.876 |
| DaEn+De+GCN+EG | 0.836 |

In the first experiment, FFTransformer (En only) (Gorishniy et al., 2021) served as the baseline, yielding an RMSE of 1.158. To extend the model to the temporal dimension, we added a transformer decoder (De) and used a learnable tensor, matching the size of the node embeddings, as the decoding target. This reduced the RMSE to 1.071, indicating that temporal modeling improves prediction accuracy.

Next, we introduced a spatiotemporal GCN by constructing a K-nearest neighbor (KNN) graph based on pixel coordinates, enabling spatial aggregation of node embeddings across both spatial and temporal dimensions. This integration of spatiotemporal information further reduced the RMSE to 0.893, highlighting the importance of modeling spatial heterogeneity using graphs and GCNs.

In the following step, we enhanced the graph by switching from a pixel-based coordinate system to a spherical coordinate system and applying our Enhanced Graph (EG) method, which incorporates Gaussian similarity-based edge weighting. This improvement resulted in an RMSE of 0.876, demonstrating the effectiveness of refining graph construction.

Finally, we introduced the Dual Attention Encoder (DaEn) to capture both temporal and feature dependencies by applying dual self-attention mechanisms. This final addition led to the most significant improvement, reducing the RMSE to 0.836.

## 4.5 VISUALIZATION

In the final experiment, we visualize the graph node embeddings to investigate whether our end-to-end learning method captures generalizable patterns from the data. We reduce the dimensionality of the node embeddings at each time step using PCA and visualize them according to their spatial locations. The results reflect the similarity or divergence in the relationships between independent and dependent variables across different locations (i.e., points closer together after PCA likely indicate similar relationships between the variables). The results are presented in Figure 5.

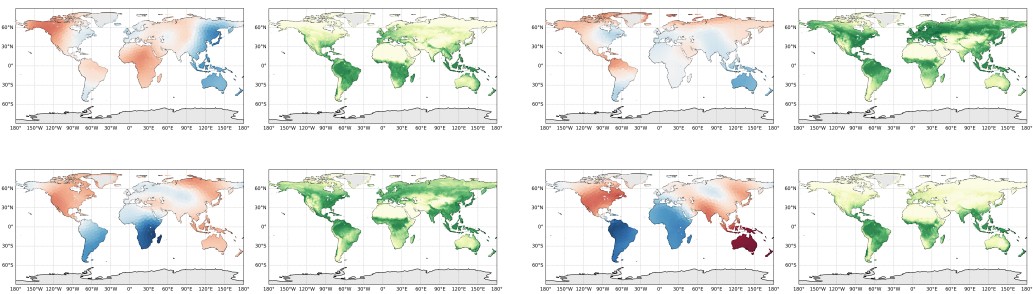

Figure 5: Spatiotemporal heterogeneity weights and total primary productivity predictions for weeks 10, 20, 30, and 40

As shown in Figure 5, our model's graph node embeddings reveal several intriguing spatial patterns. For instance, in the Week 20 visualization, the Middle East, the Sahara Desert, and central Australia

exhibit similar embedding patterns, which is consistent with these regions all containing large desert areas. Conversely, tropical areas like the Amazon rainforest, as well as subtropical regions such as southern China, display similar patterns, likely due to shared influences on vegetation growth in these climates. Although further exploration into the interpretation of graph node embeddings is warranted, these preliminary results already demonstrate that our end-to-end method effectively learns spatiotemporal heterogeneity patterns with a degree of interpretability.

## 5 CONCLUSION

In this paper, we addressed the problem of modeling spatiotemporal heterogeneity in Earth science by designing an end-to-end learning approach that captures both global patterns and localized variations. Our method was validated on large-scale climate and vegetation data, where it outperformed existing models. We draw the following conclusions: (1) the end-to-end design effectively learns common global features and improves performance compared to traditional methods, (2) ablation studies show that learning locally shared spatiotemporal heterogeneity conditions reduces overfitting, and (3) graph node embedding analysis indicates our approach can capture continuous spatiotemporal heterogeneity, providing a degree of interpretability.

Looking ahead, this work primarily demonstrates the feasibility of end-to-end fitting of mappings with spatial heterogeneity, but several aspects remain to be explored. In future research, we aim to investigate whether improving the graph construction can further optimize the modeling of spatial differences and plan to explore deeper interpretability of graph node embeddings. Additionally, while building large-scale Earth science benchmarks is resource-intensive, we will continue refining these benchmarks to better evaluate the effectiveness of future methods.

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

# A  APPENDIX

## A.1  BACKGROUND

**What is Gross Primary Productivity (GPP):** Gross Primary Productivity (GPP) measures the amount of Carbon Dioxide ($CO_2$) that plants absorb from the atmosphere and convert into biomass through photosynthesis. In simple terms, GPP represents how much energy plants capture from sunlight to support growth. GPP is driven by several environmental and biological factors, each of which plays a key role in plant growth: Solar radiation, Temperature, Water availability, Nutrient availability, $CO_2$ concentration and Vegetation type and biodiversity. These factors interact in complex ways, and their influence on GPP can vary across different geographical regions.

**Why is a Spatially-Aware Model Necessary for GPP Prediction:** Although many factors influencing GPP are measurable, we are not always able to fully observe all of them. This incomplete measurement means that the unobserved variables often vary across space in non-stationary ways. As a result, the relationship between the observed variables (e.g., temperature, radiation) and GPP also changes with spatial location. Therefore, a spatially-aware model is essential to capture these location-dependent relationships and make accurate predictions.

## A.2  HYPERPARAMETER ABLATION EXPERIMENT

We performed an extensive hyperparameter ablation study to investigate the influence of three critical hyperparameters on our method's performance: the number of spatial nodes in the **Spatiotemporal Conditional Graph** (STCG), the number of neighbors each node considers when constructing the graph, and the number of embedding channels for each node in the graph. The results, reported in terms of RMSE, are presented in Tables 3, 4, and 5.

| N_Clusters | 375 | 750 | 1500 | 3000 | 6000 | 12000 |
|---|---|---|---|---|---|---|
| **RMSE** | 0.886 | 0.876 | 0.886 | 0.895 | 0.889 | 0.947 |

Table 3: RMSE for different numbers of spatial nodes in STCG (N_Clusters).

Table 3 shows the effect of varying the number of spatial nodes (N_Clusters) in the Spatiotemporal Conditional Graph (STCG). As observed, the model performs optimally when 750 nodes are used, achieving the lowest RMSE of 0.876. Increasing or decreasing the number of spatial nodes beyond this value leads to a slight degradation in performance. For instance, with 375 nodes, the RMSE increases to 0.886, while using 12000 nodes yields the worst RMSE of 0.947. This suggests that an optimal number of spatial nodes balances the model's capacity to capture spatial variability while preventing overfitting or underfitting the spatial structure of the data.

| N_Neighbor | 10 | 20 | 30 | 50 |
|---|---|---|---|---|
| **RMSE** | 0.902 | 0.891 | 0.876 | 0.878 |

Table 4: RMSE for different numbers of neighboring nodes (N_Neighbor) considered in graph construction.

Table 4 summarizes the results of varying the number of neighboring nodes (N_Neighbor) considered for each spatial node in the graph. As shown, the model achieves its best performance with 30 neighbors, reaching an RMSE of 0.876. When fewer neighbors are used (e.g., 10 neighbors), the performance degrades slightly to an RMSE of 0.902, indicating that insufficient spatial information is being aggregated. On the other hand, using more neighbors, such as 50, also increases the RMSE to 0.878, potentially due to over-smoothing effects, where too much spatial information dilutes the model's ability to capture local spatial heterogeneity.

| Channel of Node Embedding | 32 | 64 | 96 | 128 |
|---|---|---|---|---|
| **RMSE** | 1.15 | 0.910 | 0.876 | 2.16 |

Table 5: RMSE for different numbers of embedding channels (Channel of Node Embedding).

In Table 5, we investigate the effect of varying the number of embedding channels per node in the graph. The optimal configuration is achieved when 96 embedding channels are used, yielding an RMSE of 0.876. Notably, reducing the number of channels to 32 results in a significant performance drop, with an RMSE of 1.15. Similarly, increasing the number of channels to 128 leads to an even worse result, with an RMSE of 2.16. These findings suggest that 96 channels strike the right balance between model expressiveness and overfitting, providing enough capacity to represent node features without over-complicating the model's representation.

In summary, the hyperparameter ablation study highlights that the best configuration for our method is achieved with 750 spatial nodes (N_Clusters), 30 neighbors per node (N_Neighbor), and 96 embedding channels per node. These settings provide the most accurate results, balancing model complexity and the ability to capture spatiotemporal dependencies effectively. Over-adjusting any of these hyperparameters either underutilizes or overwhelms the model's capacity to represent the data.

### A.3 VARIABLES USED IN THE MODEL

In our model, we utilize a diverse set of variables capturing key environmental and climate-related factors. These variables are categorized into three main groups: temperature-related features, land cover types, and other environmental features.

The temperature-related features include:

- `temperature_2m`: Temperature at 2 meters above ground level.
- `temperature_2m_max`: Maximum temperature at 2 meters above ground.
- `temperature_2m_min`: Minimum temperature at 2 meters above ground.
- `dewpoint_temperature_2m`: Dew point temperature at 2 meters above ground.
- `skin_temperature`: Temperature at the surface of the Earth.
- `soil_temperature_level_1` to `soil_temperature_level_4`: Soil temperature at four different depth levels.

Additionally, we include 17 land cover types:

- `land_cover_type_0` to `land_cover_type_16`: These represent various land cover categories, capturing different types of terrain and vegetation.

The model also incorporates other environmental variables, including:

- `evaporation_from_bare_soil_sum`: Total evaporation from bare soil.
- `evaporation_from_open_water_surfaces_excluding_oceans_sum`: Total evaporation from open water surfaces excluding oceans.
- `evaporation_from_the_top_of_canopy_max`, `min`, and `sum`: Maximum, minimum, and total evaporation from the top of the canopy.
- `evaporation_from_vegetation_transpiration_max`, `min`, and `sum`: Maximum, minimum, and total transpiration from vegetation.
- `total_evaporation_sum`: Total overall evaporation.
- `leaf_area_index_high_vegetation` and `low_vegetation`: Leaf area index for high and low vegetation.
- `surface_net_solar_radiation_sum`: Total surface net solar radiation.
- `volumetric_soil_water_layer_1` to `layer_4`: Volumetric soil water content in four soil layers.
- `total_precipitation_sum`: Total precipitation accumulated.

