# OpenReview forum: "Modeling Spatiotemporal Heterogeneity in Earth Science Machine Learning: An End-to-End Approach"
_ICLR.cc/2025/Conference — ICLR 2025 Conference Withdrawn Submission_

### Official Review · Reviewer_dtfC · 2024-10-18

**Soundness:** 1
**Presentation:** 2
**Contribution:** 2
**Rating:** 3
**Confidence:** 5

**Summary:**

In this paper, the authors propose a conditional generative model with local parameter sharing to replace explicit geographic and temporal weighting models. This reduces the risk of overfitting caused by dense spatio-temporal weights and improves performance in Earth Sciences Forecasting.

**Strengths:**

+ 1. The goal of this paper is to address an important real-world problem: achieving spatio-temporal forecasting from Earth science-related data. This research problem is meaningful because spatio-temporal forecasting in Earth sciences help humans understand the dynamic changes in the Earth's systems and can be applied to important areas such as agriculture and social activities.

---

+ 2. The core idea of the paper is to avoid the overfitting problem caused by global dense spatio-temporal weights by proposing a conditional generative model with local parameter sharing.

---

+ 3. The authors' writing is clear and easy to understand.

**Weaknesses:**

+ 1. In line 175, the authors project geographic coordinates onto a three-dimensional unit sphere. For a large number of nodes, this could lead to computational overhead. Additionally, this projection assumes the Earth is a perfect sphere, but in reality, the Earth is a slightly flattened ellipsoid. Therefore, spherical projection may introduce some inaccuracies in certain cases. Moreover, this method cannot effectively capture true geodesic distances.

---

+ 2. In line 185, the authors use the node2vec embedding method to calculate initial node representations. From my past experience, this algorithm has a certain time cost, especially for a large number of nodes. I remain skeptical about its time complexity, and the use of node2vec initialization seems to contradict the end-to-end approach mentioned by the authors in the abstract.

---

+ 3. In Section 3.2, the entire spatio-temporal conditional encoding process is not novel, as it can be found in various spatio-temporal graph learning methods [1, 2]. However, the authors have entirely neglected to cite relevant work.

---

+ 4. In Section 3.3, the table representation aggregation process is also not novel, as it essentially amounts to a simple application of linear attention. Considering there are many alternatives[3, 4], the authors have not explained why they chose this approach.

---

+ 5. In line 266, the details of the decoder are completely missing. If space constraints were an issue, they should have been discussed in the appendix. However, it is clear the authors are not restricted by space here.

---

+ 6. There is a lack of empirical experiments related to efficiency.

---
[1] Wu, Zonghan, et al. "Graph wavenet for deep spatial-temporal graph modeling." IJCAI 2019.
[2] Bai, Lei, et al. "Adaptive graph convolutional recurrent network for traffic forecasting." NIPS 2020.
[3] Shen, Zhuoran, et al. "Efficient attention: Attention with linear complexities."  CVPR 2021.
[4] Wang, Sinong, et al. "Linformer: Self-attention with linear complexity." arXiv 2020.

**Questions:**

+ 1. In line 42, the statement that the spatial distribution of missing variables being non-stationary implies that the relationship between the remaining features and the target variable changes with spatial location seems logically flawed to me. In my view, the spatial distribution of variables with spatiotemporal associations is generally non-stationary, regardless of whether data is missing or not.

---

+ 2. Why do the authors not summarize their main contributions at the end of the introduction, but instead scatter them throughout the paper? This makes it difficult to quickly identify their core contributions.

---

+ 3. I don’t quite understand Figure 3. Why is there such a large difference between GNNWR and GTWR compared to the other two figures? Shouldn’t spatial heterogeneity result in clear color separations across different regions? From this perspective, the GWR figure seems to capture this better.

---

+ 4. What is the time complexity of the proposed method?

---

> ### Author Response · Authors · 2024-11-13
> **Reply on Question -1**
>
> Thank you for this insightful comment. We would like to clarify that we are actually discussing **two different types of non-stationarity**: the spatial distribution of variables themselves, and the relationship between predictor and response variables.
>
> We fully agree with your observation that spatiotemporal data inherently exhibits non-stationary spatial distributions. In fact, this spatial non-stationarity of variables is one of the fundamental reasons why the relationships between predictors and response variables can also become non-stationary across space.
>
> Let us elaborate using our specific case of plant carbon accumulation. In an ideal scenario where we have complete information about all relevant variables (including plant species, phenological states, meteorological conditions, and soil properties), the underlying biophysical function governing plant carbon accumulation would be spatially and temporally invariant. This is because the fundamental biological processes respond to immediate environmental conditions rather than absolute spatial location or time.
>
> > For instance, plants can be grown in greenhouses anywhere on Earth, given the right conditions - the biological processes depend on the immediate environment rather than geographical location.
>
> However, when some variables are missing from our dataset, the spatial non-stationarity of these unobserved variables introduces apparent spatial dependence in the relationships between the observed variables (x') and the target variable (y).
>
> > Consider rice cultivation within the same region: even with similar temperature and humidity conditions, rice growth patterns may vary significantly across different locations due to unobserved soil properties (e.g., nutrient content, soil structure). The spatial non-stationarity of these unmeasured soil characteristics leads to spatially varying relationships between observed variables (like temperature and humidity) and rice growth, as the plants' response to these factors depends on the underlying soil conditions.
>
> This transforms our modeling objective from learning a spatially invariant function F(x) to learning a spatially-dependent function F(x', loc, t), where x' represents the observed subset of the full variable set x. Therefore, the non-stationarity of missing variables does indeed lead to spatially varying relationships between the remaining predictors and the target variable, not as a logical flaw but as a mathematical consequence of incomplete information.

---

> > ### Comment · Reviewer_dtfC · 2024-11-13
> >
> > Alright, I agree with this part of the response, it is very straightforward and engaging.

---

> > > ### Comment · Reviewer_dtfC · 2024-11-13
> > > **Summary**
> > >
> > > I have reviewed all the remaining responses, and overall, the authors have admitted to most of the issues I pointed out. Although they promise to make improvements in the future, there are too many aspects that need revision. While I admire the authors' hard work, I believe that now is not the right time to publish this still immature paper.

---

> > > > ### Author Response · Authors · 2024-11-13
> > > >
> > > > We appreciate the reviewer's recognition of our core work and their constructive suggestions for improvement. We will submit a revised version addressing all concerns before the rebuttal period ends.

---

> > > > > ### Comment · Reviewer_dtfC · 2024-11-13
> > > > >
> > > > > I am pleased to see that you are willing to put in the effort to refine this paper, but the acknowledged issues are inherent. Completely changing many method setups would be unfair to other authors. Therefore, I would like to maintain the original score and increase the confidence level. I wish you good luck.

---

> ### Author Response · Authors · 2024-11-13
> **Reply on Questions -2.3.4**
>
> Thank you for your valuable feedback. Let me address your questions:
>
> **Q2:** We acknowledge your point about the presentation of contributions. In the revised version, we will add a clear summary of our main contributions at the end of the introduction section to help readers quickly grasp the key advances of our work.
>
> **Q3:** Yes, the spatial heterogeneity weights learned by **GWR** align better with our theoretical predictions. The differences in spatial weights among **GTWR**, **GNNWR**, and **GWR** arise from distinct factors:
>
> For **GTWR**: While it extends GWR by learning temporal variations in spatial heterogeneity, the results appear suboptimal. We suspect this is due to GTWR's automatically estimated temporal bandwidth being too large, which introduces temporally irrelevant data during training.
>
> For **GNNWR**: The situation is different as it fits spatial weights independently for each temporal dimension. We hypothesize that the less satisfactory results may be due to overfitting on the training set, given GNNWR's significantly higher fitting capacity compared to both GTWR and GWR.
>
> **Q4:** Regarding time complexity, since our spatiotemporal conditional graph has a relatively small number of nodes (**750 nodes** with **50 edges** each), the graph construction and weight initialization process typically completes within **5 minutes**. The entire training process takes approximately **400 minutes** (**20 minutes/epoch**) on a single **RTX4090** GPU. For inference, predicting results for **64,624** spatial locations across weeks 1-44 of 2020 takes about **2 minutes**.

---

> ### Author Response · Authors · 2024-11-13
> **Reply on Weaknesses - 1**
>
> Our primary objective was to construct a graph with cyclic structure, so we initially built this graph under an idealized spherical model to calculate initial distance weights, specifically to validate the effectiveness of our cyclic graph construction approach. Regarding the computational complexity concerns, we do not treat each spatial point as an individual graph node. Instead, we employ clustering to merge adjacent spatial points, ultimately constructing the graph on 750 spatial points. This approach maintains reasonable computational demands, typically completing within a few minutes.
>
> Finally, we acknowledge the reviewer's suggestion about using an ellipsoidal coordinate system is valid, and we plan to explore this approach and update our paper accordingly.

---

> ### Author Response · Authors · 2024-11-13
> **Reply on Weaknesses - 2**
>
> We appreciate the reviewer's concerns regarding computational complexity. Our implementation employs a local weight sharing strategy with 750 cluster centers for graph construction. This means that spatial points within small regions generated by K-means share the same output embedding for decoding. With only 750 nodes and approximately 50 edges per node, the node2vec computation does not introduce significant computational overhead and typically completes within minutes.
>
> Concerning the end-to-end nature of our approach: While we understand the reviewer's concern about using node2vec initialization, we would like to clarify that:
>
> - node2vec+RoPE is only used to calculate the initial node embeddings to ensure training stability
> - Both the node embeddings and edge weights are defined as `nn.Parameters` with `require_grad=True`
> - These parameters are further optimized during backpropagation
> - The initialization merely provides a starting point, after which the model learns and adjusts these parameters end-to-end through gradient descent
>
> This implementation allows us to maintain the benefits of end-to-end learning while leveraging node2vec's structural insights for better initialization, ultimately contributing to more stable and effective training.
>
> We welcome any further discussion regarding the learning of these parameters if the reviewer has additional concerns.

---

> ### Author Response · Authors · 2024-11-13
> **Reply on Weakness - 3.4.5.6**
>
> **W3,W4:** While we acknowledge the need to include additional citations regarding spatio-temporal weight initialization and linear attention in our revised version, it's important to note that **our work addresses a fundamentally different problem from traditional spatio-temporal prediction tasks**. Our focus is on **tabular machine learning with spatio-temporal characteristics**, where each row in the table represents data from different locations and timestamps. This is distinct from conventional spatio-temporal learning methods that typically deal with grid-like or graph-structured data.
>
> The core contribution of our paper lies in how we **reformulate the challenging problem of modeling non-stationary relationships in tabular Earth science data**, where the associations between features and targets vary across both space and time dimensions. While we utilize some existing technical components as implementation tools, our **main innovation is in the novel problem formulation and solution framework** for handling such complex tabular data structures.
>
> **W5:** Regarding the decoder details, we mentioned in the paper that we used the **standard PyTorch transformer decoder implementation**, which is a widely-used module in the field. Given its widespread adoption and standardization, we didn't see the need for extensive explanation in the main text.
>
> **W6:** As for the efficiency-related experiments, please refer to our response to Q4.

---

### Official Review · Reviewer_DTJH · 2024-10-28

**Soundness:** 2
**Presentation:** 3
**Contribution:** 1
**Rating:** 3
**Confidence:** 4

**Summary:**

This paper proposes a graph neural network-based method for modeling spatiotemporal data. In the experiments, the proposed method achieved the better performance than the existing tabular machine learning, tabular deep learning methods, and geographically weighted regression.

**Strengths:**

Proposed a new method for earth science.
Experiments with global data.

**Weaknesses:**

The technical contribution of the proposed method is incremental. There have been proposed many deep learning methods for spatiotemporal modeling, such as graph neural networks and Transformers. The proposed method is a combination of graph neural networks and self-attention models. The experimental results are not convincing. Comparison with deep learning methods for spatiotemporal data is needed to clarify the effectiveness of the proposed method.
Although the authors describe that the proposed method is an end-to-end approach, there are many components that are not trained in an end-to-end fashion; i.e., clustering of spatial regions, graph construction, initial node embedding, and edge weight calculations.

**Questions:**

What is the main contributions of this paper compared with the existing deep learning methods for spatiotemporal data?

---

> ### Author Response · Authors · 2024-11-13
> **Our Task is a Tabular Machine Learning Problem, Not a Traditional Spatiotemporal Prediction Problem**
>
> **We appreciate the reviewer's comments but would like to clarify a potential misunderstanding about the nature of our research problem. The GPP prediction task we address is fundamentally different from traditional spatiotemporal prediction problems like traffic or crowd flow forecasting.**
>
> Our task has several unique characteristics:
>
> 1. **Our Task is a Tabular Machine Learning Problem, Not a Traditional Spatiotemporal Prediction Problem**:
> Our task is essentially a tabular machine learning problem where we aim to fit the relationship between meteorological data and GPP. **Most importantly, we can achieve reasonable accuracy using traditional tabular machine learning methods even if we ignore all spatial and temporal conditions.** This fundamentally distinguishes our problem from conventional spatiotemporal prediction tasks like traffic flow forecasting.
>
> 2. **Temporal Dependency**:
> Unlike traffic flow prediction where strong temporal dependencies exist between consecutive time steps, the temporal correlation in our GPP prediction is relatively weak. GPP values are primarily determined by current meteorological conditions rather than their historical values. This makes it inappropriate to compare our method with those specialized deep learning approaches designed for strong temporal dependency problems.
>
> 3. **Research Objective**:
> Our objective is to explore how to model non-stationary relationships between variables in space and time when key independent variables are missing in a tabular machine learning context. This is fundamentally different from traditional spatiotemporal prediction tasks, as we are trying to capture how missing variables affect the relationship between observed features and target variables across different spatial and temporal contexts.
>
> 4. **Our Key Contributions**:
>    * We innovatively introduce spatial-temporal conditions into a tabular learning problem where data points are inherently assumed to be independent
>    * We design a weight-sharing mechanism to prevent local overfitting of spatial-temporal conditions
>    * Our end-to-end learning framework effectively balances the influence of meteorological features and spatial-temporal conditions
>    * This design specifically addresses the challenge of incorporating weak spatial-temporal correlations without overwhelming the primary meteorological feature relationships
>
> Therefore, comparing our method with specialized deep learning approaches designed for strong temporal dependency problems would not be appropriate, as these methods are optimized for fundamentally different problem characteristics.
>
> We suggest that a more appropriate evaluation would be to compare our method with other tabular machine learning approaches, demonstrating how our spatial-temporal-aware framework improves upon traditional methods in this specific context.

---

> > ### Comment · Reviewer_dtfC · 2024-11-13
> > **Agree with the reviewer DTJH's opinion**
> >
> > Why do the authors think their method is an end-to-end framework, it is obviously not! If the authors are not clear about even the basic definitions and concepts, they should humbly admit it.

---

> > > ### Author Response · Authors · 2024-11-13
> > >
> > > Thank you for your comment. We would like to clarify our definition of end-to-end learning: In our framework, except for the graph connectivity which is predefined, all other components are learnable parameters, including:
> > >
> > > Encoder module
> > > Decoder module
> > > Node embeddings
> > > Edge weights
> > > All these components have require_grad=True and are jointly optimized under the same loss function. If you believe there are other non-learnable parts, please specify them, as this would help us better understand your concerns and improve our work.

---

> > > > ### Comment · Reviewer_dtfC · 2024-11-13
> > > >
> > > > Well, the authors acknowledge that they first need to perform a predefined graph processing operation and a pre-training step for the nodes (implemented by the node2vec). The former can be seen as data engineering, while the latter requires separate training, although it might be lightweight. Based on these, the authors then train the actual model. However, this cannot truly be called end-to-end training, because the learning of the latter and the former is not coupled (of course, I know this is a common operational paradigm from 2015-2019, conceptually similar to pre-training + fine-tuning). I really can't understand why they would call their framework end-to-end.

---

> ### Author Response · Authors · 2024-11-13
>
> Thank you for your thoughtful comment regarding the end-to-end nature of our framework. We would like to clarify that Node2Vec is not actually essential to our method. We had verified that initializing node embeddings with a standard normal distribution plus RoPE positional encoding in earlier experiment that could achieve similar accuracy levels as reported in our paper, albeit requiring longer training time (approximately 18 hours).
>
> If you maintain that using Node2Vec constitutes a form of pre-training that violates true end-to-end training principles, we are willing to revise our paper to use the standard initialization approach without Node2Vec across all experiments. We believe this modification would align with your definition of end-to-end training.
>
> We appreciate your feedback as it helps us improve the clarity and technical accuracy of our work.

---

### Official Review · Reviewer_GPTX · 2024-10-31

**Soundness:** 2
**Presentation:** 2
**Contribution:** 2
**Rating:** 3
**Confidence:** 3

**Summary:**

This paper introduces a novel end-to-end framework for modeling spatiotemporal heterogeneity in Earth science data using a conditional generative approach. The authors propose a model that leverages GNNs and transformer-based encoding to handle spatial and temporal dependencies simultaneously. Through the Climate2GPP dataset, the method demonstrates superior performance over established techniques such as GWR, LightGBM, and TabNet in predicting GPP.

**Strengths:**

The topic is interesting. The visualization of results is clear and well-designed. Applying the proposed method to a large-scale dataset is both challenging and impactful.

**Weaknesses:**

1.	The problem statement is not clearly defined. It would be beneficial to add a dedicated subsection at the beginning to outline the problem statement, as well as explicitly describe the input features, target variables, and intended outputs.

2.	For the spatial heterogeneity problem, it does not sufficiently discuss existing solutions, such as using region division strategy [1], meta-learning [2] or considering dynamic environmental configurations to capture complex variable relationships [3]. A brief comparison with these methods would contextualize the study.

Reference:

[1] Deep transfer learning for intelligent cellular traffic prediction based on cross-domain big data. IEEE Journal on Selected Areas in Communications. 2019

[2] Urban traffic prediction from spatio-temporal data using deep meta learning, KDD 2019

[3] Long-term Forecasting with TiDE: Time-series Dense Encoder. Transactions on Machine Learning Research. 2023

**Questions:**

1.	Will the dataset and code used in this study be made publicly available?

2.	The study uses data from 2001-2019 for training and data from 2020 for testing. Given the significant environmental and impacts of COVID-19 in 2020, is it possible that this anomaly could affect the model’s predictions?

---

### Official Review · Reviewer_WNLQ · 2024-11-01

**Soundness:** 3
**Presentation:** 3
**Contribution:** 3
**Rating:** 6
**Confidence:** 3

**Summary:**

This paper proposes a novel graph neural network-based method to model spatiotemporal heterogeneity in Earth science data. It incorporates a Spatiotemporal Conditional Graph (STCG) to integrate both spatial and temporal data, capturing environmental changes dynamically. A dual attention mechanism within a transformer architecture enhances the model's ability to handle complex dependencies across time and features. The effectiveness of the proposed method is demonstrated using real-world data sets.

**Strengths:**

- The entire modeling process, from data preprocessing to final prediction, is optimized end-to-end. This unified approach ensures that the model learns generalized features across the entire dataset, which helps in enhancing predictive accuracy.
- The manuscript is well-written.
- Although the proposed approach is simple and a combination of several elements, the way it is combined is reasonable and practically useful.
- The effectiveness of the proposed method is demonstrated using real-world data sets.

**Weaknesses:**

- There are some unclear points in the formulation.
- There is no discussion of computational complexity.
- More detailed study of experimental results would be helpful.

**Questions:**

- I did not understand how the cluster center C in section 3.1 is used in the architecture. Could you please explain in detail?
- The edge weight is calculated based on Euclidean distance, but is it necessary to use other geometric distances when considering a spherical surface like the earth?
- Is it possible to do a detailed analysis of the prediction results? For example, do the errors differ from region to region? Are local and global dynamics really captured? Are there any areas where the error is locally larger?

---

### Note · Authors · 2024-11-13

I have read and agree with the venue's withdrawal policy on behalf of myself and my co-authors.